# Honey as an Adjuvant in the Treatment of COVID-19 Infection: A Review

Sónia Soares [1,*], Mélina Bornet [1], Clara Grosso [1], Maria João Ramalhosa [1], Irene Gouvinhas [2], Juliana Garcia [2,3], Francisca Rodrigues [1] and Cristina Delerue-Matos [1]

[1] REQUIMTE/LAQV, Instituto Superior de Engenharia do Porto, Rua Dr. António Bernardino de Almeida, 4249-015 Porto, Portugal; melina.brnt08@gmail.com (M.B.); claragrosso@graq.isep.ipp.pt (C.G.); mjr@isep.ipp.pt (M.J.R.); francisca.rodrigues@graq.isep.ipp.pt (F.R.); cmm@isep.ipp.pt (C.D.-M.)

[2] Centre for the Research and Technology of Agro-Environmental and Biological Sciences (CITAB)/Institute for Innovation, Capacity Building and Sustainability of Agri-Food Production (Inov4Agro), Universidade de Trás-os-Montes e Alto Douro, Quinta de Prados, 5000-801 Vila Real, Portugal; irenegouvinhas22@hotmail.com (I.G.); garciaju1987@gmail.com (J.G.)

[3] AquaValor—Centro de Valorização e Transferência de Tecnologia da Água—Associação, Rua Dr. Júlio Martins n° 1, 5400-342 Chaves, Portugal

[*] Correspondence: sonia.soares@graq.isep.ipp.pt

**Abstract:** Since ancestor times, honey has been used to promote human health due to its medicinal, and nutritious properties, mainly due to bioactive compounds present, such as phenolic compounds. The emergence of COVID-19, caused by the SARS-CoV-2 virus, led to the pursuit of solutions for the treatment of symptoms and/or disease. Honey has proven to be effective against viral infections, principally due to its potential antioxidant and anti-inflammatory activities that attenuate oxidative damage induced by pathogens, and by improving the immune system. Therefore, the aim of this review is to overview the abilities of honey to attenuate different COVID-19 symptoms, highlighting the mechanisms associated with these actions and relating the with the different bioactive compounds present. A brief, detailed approach to SARS-CoV-2 mechanism of action is first overviewed to allow readers a deep understanding. Additionally, the compounds and beneficial properties of honey, and its previously application in other similar diseases, are detailed in depth. Despite the already reported efficacy of honey against different viruses and their complications, further studies are urgently needed to explain the molecular mechanisms of activity against COVID-19 and, most importantly, clinical trials enrolling COVID-19 patients.

**Keywords:** honey; COVID-19; natural products; biological properties; antiviral activity

## 1. Introduction

The value of honey recognized all over the world. Several studies have been performed in different fields concerning honey and its related products. These natural products have effects on the environment, biodiversity, and human health assurance. The role of honey in health and wellbeing is well recognized, with its medicinal and nutritious performance being extremely explored over human history. In recent years, concerns about the environment and the preservation of the planet have increased. This had led to a search for natural products as alternatives to processed foods and chemical foodstuffs. In this sense, honey, and its related products, such as bee pollen, beeswax, beebread, and propolis, are natural products favored by consumers and industries. Based on their biological properties, these products have been selected for a wide range of applications, and as ingredient, or single products, to address old and new problems. Honey's biological, nutraceutical and medicinal properties have been scientifically proven, especially regarding antioxidant, anti-inflammatory, antibacterial and antidiabetic activities, as well as for respiratory, gastrointestinal, cardiovascular, and nervous system protective effects [1–3]. The antibacterial

activity of honey is one of its most valued properties. In fact, honey is increasingly used in its pure form, or in mixtures with drugs, for the treatment of infections, burns and wounds. Its antibacterial power is mainly attributed to its honey high osmolarity, acidity (low pH), and content of hydrogen peroxide ($H_2O_2$) and non-peroxide components [4,5]. Consequently, different studies have been performed increasingly involving honey and related products, aiming to understand and prove the real health values of this natural product [6,7]. Despite the high nutritional and medical value attributed to the generality of honey products, there are differences among them that may result in different attributes and properties. Honey can be classified as blossom or nectar honey: the first one is produced by honeybees from the nectar of plants, while the second result from secretions of living parts of plants or excretions of plant-sucking insects on the living parts of plants [3,8]. Additionally, nectar honey can be classified as unifloral or multifloral, depending on its predominant production from a single, or from several plant species, respectively. Unifloral honey is considered high-quality honey, since it can attain not only specific flavor and organoleptic properties, but also specific biological properties. Despite its essential composition of water and sugars (mainly fructose and glucose), honey also contains other minor compounds, namely vitamins, minerals, enzymes, free amino acids, and numerous volatile compounds [3,9,10]. These compounds are responsible for conferring specific/individual organoleptic, nutritional, and biological properties, depending on the botanical origin, geographic area, season, and technology used for honey extraction, as well as storage conditions [3]. Tables 1–3 summarize the most relevant bioactive compounds available in honey from different geographical origins.

As shown, the amount of bioactive compounds can differ in honeys from the same country, or in countries in close geographical proximity. These minor compounds are normally used to differentiate honeys by botanical and geographical origins, as well as to define their quality [3,10]. Since biological properties depend on the type of plants visited by bees, honey attains different preventive and therapeutic properties according to the flora surrounding beehives.

In traditional medicine, honey is used for the prevention and treatment of some disease conditions, such as asthma, throat infections, tuberculosis, eye diseases, thirst, hiccups, fatigue, dizziness, hepatitis, constipation, worm infestation, piles, eczema, healing of ulcers, and wounds [2]. However, few studies are available proving the scientific support and effectiveness of honey for medical purposes [2]. Nevertheless, new viruses and disease emergence have contributed to a new trend based on the use of natural products with promising biological properties.

**Table 1.** Sugar (g/100 g), organic acid (mg/kg) and amino acid (mg/kg) contents of honey from different geographical origins.

| Geographical Origin | Sugars (g/100 g) | | | | | | | Organic Acids (mg/kg) | | | | | | | | | Amino Acids (mg/kg) | Ref. |
|---|---|---|---|---|---|---|---|---|---|---|---|---|---|---|---|---|---|---|
| | Glucose | Fructose | Sucrose | Trehalose | Melezitose | Turanose | Maltose | Gluconic Acid | Tartaric Acid | Malic Acid | Citric Acid | Succinic Acid | Quinic Acid | Pyroglutamic Acid | Lactic Acid | Formic Acid | | |
| Brasil | 29.74–31.89 | 39.74–43.94 | - | - | - | - | - | - | - | - | - | - | - | - | - | - | - | [11] |
| Brasil | 37.7–45.4 | 50.0–59.2 | 0.7–3.9 | - | - | - | nd | - | - | - | - | - | - | - | - | - | - | [12] |
| Brasil | - | - | - | - | - | - | - | 3309.6–18,737.3 | - | <LOD–2861.2 | <LOD–1322.3 | <LOD–2292.0 | - | - | <LOD–3063.6 | <LOD–341.2 | - | [13] |
| China | - | - | - | - | - | - | - | - | - | - | - | - | - | - | - | - | 394.7–1572.9 | [14] |
| China | - | - | - | - | - | - | - | 649.0–1682.9 | - | 15.6–262.6 | 31.9–58.3 | 11.7–34.8 | - | - | - | - | - | [15] |
| China | - | - | - | - | - | - | - | - | - | 7.6–32.0 | - | 0.8–38.9 | - | - | - | 1.1–151.8 | - | [16] |
| China | 30.2–30.3 | 40.5–40.6 | 2.2–3.5 | - | - | - | - | - | - | - | - | - | - | - | - | - | 1192–1688 | [17] |
| Ecuador | 26.00–38.26 | 34.77–44.57 | 2.63–5.14 | - | - | - | - | - | - | - | 0.3–6.8 | - | - | - | 0.4–7.2 | - | - | [18] |
| Egypt | 10.63–26.54 | 4.48–50.78 | 1.34–3.59 | - | - | - | - | - | - | - | - | - | - | - | - | - | - | [19] |
| France | - | - | - | - | - | - | - | 1857–12725 | - | - | 44–434 | - | 54–1779 | 217–1962 | 125–752 | 19–1897 | - | [20] |
| Japan | - | - | - | - | - | - | - | 1337.7–6475.1 | 15 | 10.3–1724.4 | 5.7–307.4 | 7.9–91.9 | - | - | - | - | - | [15] |
| Malaysia | 12.17–40.9 | 15.03–48.44 | <0.01–7.29 | - | - | - | - | - | - | - | - | - | - | - | - | - | - | [21] |
| New Zeland | - | - | - | - | - | - | - | 1842.1–5448.9 | 2.8–7.2 | 40.0–267.6 | 6.2–288.6 | 5.1–58.6 | - | - | - | - | - | [15] |
| Poland | 26.32 | 27.6 | 0.1 | - | - | - | - | - | - | - | - | - | - | - | - | - | - | [22] |
| Portugal | 21.00–36.00 | 33.40–48.80 | 0.0–3.00 | 0.004–0.80 | 0.18–1.20 | 0.001–0.69 | 0.93–7.83 | - | - | - | - | - | - | - | - | - | - | [23] |
| Portugal | 18.1–31.1 | 25.5–45.3 | <LOQ–2.0 | <LOQ–0.77 | <LOQ–6.9 | 1.2–4.0 | <LOQ–3.0 | - | - | - | - | - | - | - | - | - | - | [24] |
| Romania | 25.7–39.1 | 34.5–41.9 | nd–0.06 | - | - | - | 1.0-3.7 | - | - | - | - | - | - | - | - | - | - | [25] |
| Spain, Romania | 23.2–38.7 | 32.9–42.3 | 0.2–2.3 | - | nd–0.2 | - | 1.2–2.9 | - | - | - | - | - | - | - | - | - | - | [26] |
| Spain | - | - | - | - | - | - | - | - | - | - | - | - | - | - | - | - | 56.08–141.65 | [27] |
| Spain | - | - | - | - | - | - | - | 2877.7–3250.3 | - | 44.3–52.2 | 78.9–89.1 | 14.4 | - | - | - | - | - | [15] |
| Spain | 27.4–32.8 | 34.6–40.8 | | - | - | - | - | - | - | 47.1–241 | - | 7.1–69.7 | - | - | <LOQ-51.7 | 27.5–150.6 | - | [28] |
| Several Countries | - | - | - | - | - | - | - | - | - | - | - | - | - | - | - | - | 127.7–1523.2 | [29] |
| Tunisia | 31.07–36.58 | 35.78–37.84 | 0.20–4.6 | - | - | - | 1.36–4.34 | - | - | - | - | - | - | - | - | - | - | [30] |

LOQ—Limit of Quantification; nd—not determined.

**Table 2.** Total proteins, enzymes, TPC, TFC, bioactive compounds and carotenoids of honey from different geographical origins.

| Geographical Origin | Total Protein (mg/g) | Enzymes | | TPC (mg GAE)/kg | TFC | Bioactive Compounds (mg/kg Honey) | | | | | | | Carotenoids (mg β-Carotene/100 g) | Ref. |
|---|---|---|---|---|---|---|---|---|---|---|---|---|---|---|
| | | Diastase (Amylase) | Invertase (Saccharase) | | | *p*-Coumaric | Gallic Acid | Caffeic Acid | Syringic Acid | Vanillic Acid | Chlorogenic Acid | Quercetin | | |
| Algeria | - | - | - | - | - | - | - | - | - | - | - | - | 0.03–0.101 | [31] |
| Algeria | - | - | - | 640–2000 | 30–280 mg/kg | - | - | - | - | - | - | - | - | [32] |
| Australia | - | - | - | 30.9–66.3 | 4160–9640 mg CE/kg | - | - | - | - | - | - | - | 1.74–6.96 | [33] |
| Brasil | - | 11.14–22.69 DN | - | - | - | 0.22–1.43 | - | nd–0.46 | - | nd–0.90 | - | nd–1.58 | - | [11] |
| Brasil | - | - | - | 73.93 | 2.03 mg/kg | - | 30.9 | nd | nd | nd | nd | - | - | [34] |
| Brasil | 2.0–5.0 | - | - | - | - | - | - | - | - | - | - | - | - | [12] |
| Canada | - | - | - | 136.87 | 6.26 mg/kg | - | 32.1 | nd | nd | nd | nd | - | - | [34] |
| China | - | - | - | 22.90–159.04 | nd–4.42 mg/kg | - | nd–66.2 | nd–23.6 | nd–4.9 | nd | nd–21.2 | - | - | [34] |
| Croatia | - | 7.5–37.3 DN | 26.4–277.9 U/kg | - | - | - | - | - | - | - | - | - | - | [35] |
| Ecuador | 0.02–0.37 | 8.33–40 DN | - | - | - | - | - | - | - | - | - | - | - | [18] |
| Egypt | 1.69–4.67 | - | - | - | - | - | - | - | - | - | - | - | - | [19] |
| Ethiopia | - | 7.64–12.5 DN | - | - | - | - | - | - | - | - | - | - | - | [36] |
| Germany | - | - | - | 105.13–135.22 | 2.72–6.83 mg/kg | - | 29.3–45.5 | nd–14.8 | nd–3.2 | nd | nd | - | - | [34] |
| Italy | - | - | - | 400–730 | 20.66–30.56 mgCE/kg | - | - | - | - | - | - | - | - | [37] |
| Italy | - | - | - | 81.29–148.75 | 1.59–12.35 mg/kg | - | 21.0–55.9 | nd–20.3 | nd–18.4 | nd | nd–19.3 | - | - | [34] |
| Malaysia | - | - | 0.27–4.94 IN | - | - | - | - | - | - | - | - | - | - | [21] |
| Portugal | - | 15.2–15.6 DN | - | 678.4–698.1 | 494.4–563.3 mg/kg | - | - | - | - | - | - | - | - | [38] |
| Portugal | - | 16.15–37.77 DN | - | 139.52–591.87 | 16.46–112.83 mg/kg | - | - | - | - | - | - | - | - | [39] |
| Romania | - | - | - | - | - | 3.6-29.1 | 0.5–0.9 | 0.2–7.5 | - | - | - | - | - | [40] |
| Spain | 0.83–0.93 | - | - | - | - | - | - | - | - | - | - | - | - | [27] |
| Spain | - | - | - | 74.57–254.03 | 2.66–41.65 mg/kg | - | 28.5–63.6 | nd–20.9 | nd–87.7 | nd–5.1 | nd–19.6 | - | - | [34] |
| Tunisia | 0.13–0.16 | - | 46.25–184.68 U/kg | 32.17–119.42 | 9.58–22.45 mg CE/kg | - | - | - | - | - | - | - | 1.16–4.72 | [30] |
| Turkey | - | - | - | 1813–60510 | 7660–28,750 mg/kg | - | nd–390 | 60–4610 | - | - | nd–1140 | nd–1430 | - | [41] |

nd—not detected; TPC—Total Phenolic Content; TFC—Total Flavonoid Content.

**Table 3.** Mineral composition and trace elements (mg/kg) of honey from different geographical origins.

| Geographical Origin | Minerals and Trace Elements (mg/kg) | | | | | | | | Ref. |
|---|---|---|---|---|---|---|---|---|---|
| | Fe | Zn | Cu | Mn | Na | K | Ca | Mg | |
| Argentina | 2.07–4.5 | 0.51–2.75 | 0.09–1.19 | 0.14–8.84 | 4.88–105.95 | 134.1–2813.3 | 1.97–18.97 | 3.01–75.38 | [42] |
| Bulgaria | 0.71–19.25 | 0.71–1.71 | 0.05–0.49 | 0.31–4.7 | 6.3–20.2 | 136–1900 | 24–94 | 8.3–48 | [43] |
| Ecuador | nd | nd | nd | nd | 9.0–23.0 | 7.0–133.0 | 15.0–31.0 | 4.0–11.0 | [13] |
| Greece | 1.03–6.30 | 0.89–1.81 | 0.14–0.52 | 0.15–1.46 | 10.3–42.0 | 391–2494 | 18.0–78.0 | 6.8–63.6 | [44] |
| Pakistan | 0.04–0.19 | - | - | 1.05–3.11 | 211.6–579.6 | 166.5–465.66 | 0.5–0.73 | - | [45] |
| Pakistan | 2.98–16.2 | 1.11–4.1 | 0.08–0.33 | 0.12–0.95 | 77.5–200 | 225–439 | 46.1–98.1 | 31.3–73.8 | [46] |
| Poland | nd–16.1 | nd–9.93 | nd–1.82 | - | 0.38–89.6 | 7.7–2612.2 | 3.3–159.2 | 0.07–19.83 | [47] |
| Slovakia | 1.02–5.14 | 0.16–1.30 | 0.045–2.01 | 0.44–15.1 | 8.49–10.3 | 0.33–3.71 | 20.3–36.6 | 12.5–65.0 | [48] |
| Tunisia | 0.83–3.54 | 0.42–2.06 | 0.12–0.34 | - | 251.34–521.22 | 172.48–976.75 | 113.85–221.07 | 37.32–78.12 | [30] |
| Turkey | <0.001–7.25 | <0.001–0.24 | <0.001–0.93 | <0.001–0.27 | 0.48–13.1 | 1.18–268 | 0.77–4.5 | - | [49] |
| Turkey | BDL–14.0 | BDL–1.98 | BDL–0.46 | BDL–0.82 | - | - | - | - | [50] |

BDL—bellow the detection limit; nd—not detected.

A recent example is COVID-19, which since December 2019, has been a world public health concern. This disease quickly proved to be highly contagious, resulting in severe symptoms, especially at the respiratory level. Caused by SARS-CoV-2, COVID-19 has been declared a considered a pandemic, and efforts at the global level have been directed towards finding and ng cures and treatments. This novel pandemic, and highly contagious disease, has become a priority. Studies directed to the understanding of its clinical symptoms began early [51]. The necessity of controlling the SARS-CoV-2 pandemic virus has mobilized the scientific and medical community to adapt treatments for existing viruses and develop new strategies and vaccines. In recent years, many herbs and plants have attained crucial importance in the pharmaceutical industry due to their use in traditional medicine based on their bioactive and biological properties, including antiviral activity. Different plant bioactive compounds, such as flavonoids, have been identified, studied, and used for the treatment of certain diseases, among then honey. Therefore, the main goal of this review is to explore the use of honey as co-adjuvant in the treatment of SARS-CoV-2 infection, highlighting its bioactive composition and biological activities. A brief overview of the viral mechanism of action, and the therapies available, are provided, aiming to highlight the advantage of honey as an active substance against SARS-CoV-2.

## 2. Methodology

Specific information on the topic was collected from the literature available from search engines such as Google Scholar, PubMed, Science Direct, Scopus, and Web of Science for retrieving published data (from 2000 to 2022) using different combination of keywords i.e., COVID-19/SARS-CoV-2, honey, mechanism of action, and immunomodulation/anti-inflammatory/antiviral, among others. The inclusion criteria were limited to full text articles on pharmacological or therapeutic approaches for COVID-19 based on in vitro, in vivo and clinical trial reports.

## 3. SARS-CoV-2 Overview and Vaccines Mechanism of Actions

The SARS-CoV-2 virus mostly targets the cells of the lower respiratory tract (due to its easy binding to epithelial cells), and replicates and migrates to the airways and alveolar epithelial cells in the lungs [52]. Extensive efforts have been made to characterize this virus through genomic sequence studies, mainly focusing on determining the viral protein structure [53–55]. Full genome-sequencing analysis has classified SARS-CoV-2 as a new member of the betacoronavirus genus, which includes SARS-CoV, MERS-CoV, and bat SARS-related coronaviruses (SARSr-CoV) [56]. Like other CoVs, its genome encodes

sixteen non-structural replicase proteins (nsp 1–16), nine accessory proteins (ORF) and four structural proteins, including spike (S), envelope (E), membrane (M), and nucleocapsid (N) proteins [56,57].

Among the structural proteins, the S protein plays an important role in virus attachment, fusion, entry, and disease pathogenesis, since it mediates viral access into the host cells by binding to a host receptor through the receptor-binding domain (RBD) in the S1 subunit and fusing the viral and host membranes through the S2 subunit 9. Thus, the S protein has been selected as a preferred target for the development of antibodies, entry inhibitors and vaccines [58,59].

In SARS-CoV-2, as in SARS-CoV, this protein binds to the receptor angiotensin-converting enzyme 2 (ACE2). Spike RBD-ACE2 binding happens not only with the human ACE2 (hACE2), but also with the ACE2 from other animals, such as pig, ferret, rhesus monkey, civet, cat, pangolin, rabbit, and dog [52]. The importance of this interaction is highlighted by the mutations on the spike protein that contribute to a different behavior of SARS-CoV-2 and cause different variants of the disease. Barton and co-workers [60] evaluated mutation effects on the spike and on the hACE2. The authors studied five common RBD mutations (K417N, K417T, N501Y, E484K, and S477N) and two common hACE2 mutations (S19P and K26R), individually and in combination with the RBD-hACE2 interaction. The mutations selected were identified in SARS-CoV-2 Alpha (B.1.1.7), Beta (B.1.351), and Gamma (P1) variants. The results demonstrated an increased affinity of the RBD-hACE2 interaction in most of the mutations studied and suggested that the N501Y and S477N mutations mainly enhance transmission by increasing binding. The K417N/T mutation facilitates immune escape as well, and the E484K mutation also improves binding. Measuring the effect of these mutations on the strength of the spike RBD-ACE2 interactions, the authors concluded that in SARS-CoV-2 Alpha (B.1.1.7), Beta (B.1.351), and Gamma (P1) variants, spike proteins bind to hACE2 more strongly than the original form, suggesting that people with these mutations are more susceptible to SARS-CoV-2.

New insights into the mechanisms of SARS-CoV-2 cell entrance will aid in the development of successful therapeutics to prevent important stages of the viral life cycle [61]. Although the mechanisms of entry for coronaviruses are not yet fully defined as other injurious effects, several pharmacological therapeutics have been applied to combat the virus. Different viral targets have been studied, such as RNA-dependent RNA polymerase, viral protease, and spike (S) glycoprotein, among others, and different vaccines strategies have been developed (Figure 1).

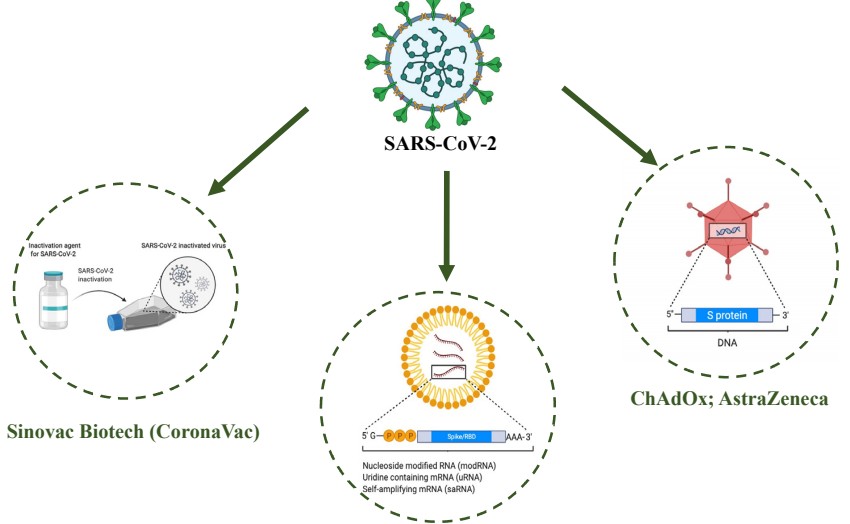

**Figure 1.** SARS-CoV-2 vaccines mechanism of actions.

### 3.1. RNA-Dependent Inhibitors

Remdesivir is a prodrug with a similar structure to adenosine that combines with nascent viral RNA and inhibits RNA-dependent RNA polymerase, terminating viral genome replication (Figure 2) [62].

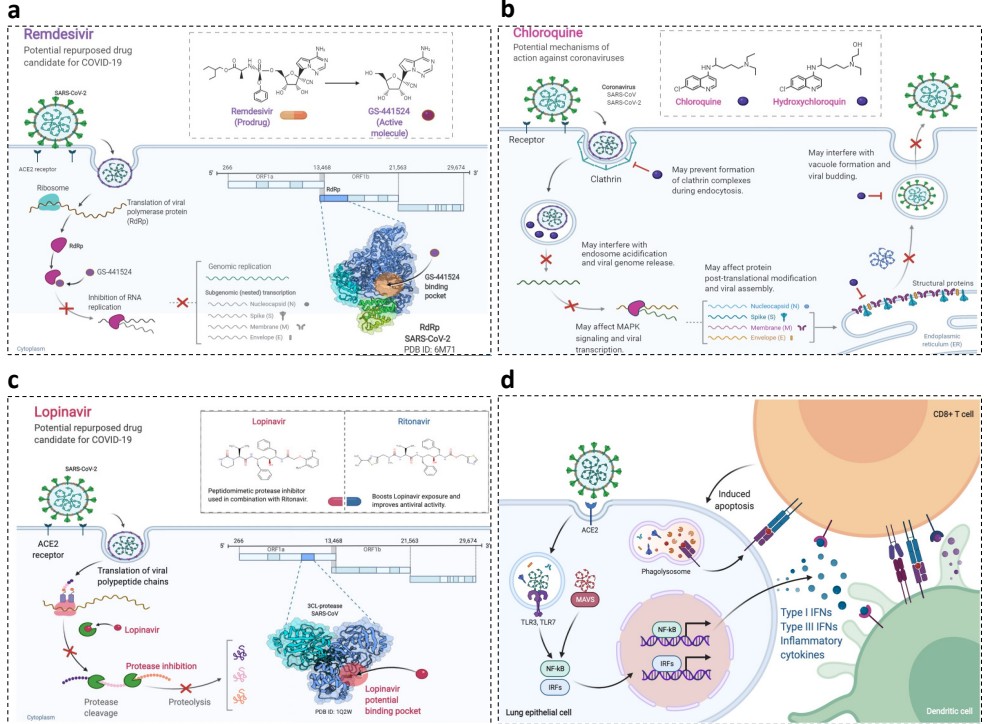

**Figure 2.** Mechanism of action of (**a**) Remdesivir, (**b**) Chloroquine, (**c**) Lopinavir and (**d**) Immunomodulators on the SARS-CoV-2 virus. Illustrations were made with BioRender.

According to randomized trials, this drug can be used to treat COVID-19, and is approved by the Food and Drug Administration (FDA). The dose should be 200 mg, followed by a daily infusion of 100 mg for 5–10 days. Two randomized, open-label, multicenter phase 3 studies are being sponsored by the Italian Medicines Agency to determine the efficacy of remdesivir as an antiviral medication versus supportive care [63,64]. Like remdesivir, favipiravir acts as an RNA-dependent RNA polymerase inhibitor, presenting a structure similar to endogenous guanine [65]. Clinical evidence for the safety and efficacy of favipiravir in COVID-19 from trials in Thailand, China, Japan and Russia, suggests benefits in the control of COVID-19, predominantly with mild to moderate disease [66].

### 3.2. Protease Inhibitors

Viral proteases are enzymes responsible for the hydrolysis of amide bonds of peptide units of viral polypeptides and protein precursors to generate mature active proteins [67]. Most viral proteases are derived from well-known eukaryotic protease families, recognizing, and cleaving diverse substrate sequences with distinct specificities [68–71]. Lopinavir/ritonavir (Figure 2) have been used clinically in SARS-CoV-infected subjects. The rhinovirus inhibitor ruprintrivir, and related compounds, have also been tested. However, in the absence of any elements of specificity, these preexisting compounds would be expected to have low potency [71]. Ivermectin was also studied for the treatment and prevention of SARS-CoV-2 by an in vitro investigation conducted by Caly et al. (2020), demonstrating that ivermectin reduced viral RNA up to 5000 times after 48 h of SARS-CoV-2 infection [72,73]. However, Popp et al. (2021) reported 14 studies (with 1678 participants) that investigated the effects of ivermectin, and only one study reported the effects of iver-

mectin in the prevention of SARS-CoV-2 infection, concluding the absence of evidence regarding the efficacy and safety of ivermectin to treat or prevent COVID-19 [74].

### 3.3. Prevention of Cell Membrane/Virus Fusion

The fusion of lipid bilayers is essential to viral infection. When the enveloped virus infects the host cell, its membrane fuses with the host-cell membrane, transferring the virus content. This fusion event is mediated by virally encoded surface glycoproteins [75]. All coronaviruses have a spike (S) glycoprotein that is embedded in the viral envelope in a trimeric form b, and is the target of some antiviral drugs.

The FDA approved an HIV protease inhibitor, nelfinavir mesylate (Viracept), that has proven to be a potent inhibitor of cell fusion caused by the SARSCoV-2 S glycoprotein [76]. On the other hand, chloroquine and hydroxychloroquine (Figure 2), an antimalarial and an anti-autoimmune drug, respectively, have shown to possess in vitro activity against the novel coronavirus SARS-CoV-2 [77,78] by fusion inhibition resulting from modulating endosomal pH [79,80].

### 3.4. Immunomodulators

Several SARS-CoV-2-infected individuals are asymptomatic, but around one-fourth of the cases of symptomatic individuals progress to severe illness, in part due to immune dysregulation [81,82]. This dysregulation leads to high inflammatory responses, including excessive production of cytokines, such as interleukin (IL)-1, IL-6, interferon (IFN)-$\alpha$, and tumor necrosis factor-$\alpha$ [83]. Corticosteroids are immunomodulatory candidates that inhibit the production of several cytokines [84,85]. Herbal derivatives, such as curcumin, sesquiterpenoids, diterpenoids or triterpenoids, also demonstrate potent inhibitory activity against COVID-19 [86–91].

### 3.5. Vaccines

The SARS-CoV-2 vaccines presently approved for therapeutic use are based on: (i) inactivated and protein subunit vaccines; (ii) viral vector vaccines, and (iii) mRNA vaccines [92]. A brief overview of each is given below.

#### 3.5.1. Inactivated and Protein Subunit Vaccines

Vaccines generated from chemically inactivated viruses are a strategy for vaccine development that contains conformationally native antigenic epitopes (Figure 1). Sinopharm and Sinovac are two of the companies that have advanced mostly in the development of this type of vaccine, which has been assessed in phase III studies and has received international approval for usage [93]. Another approach for vaccine development is the use of the viral protein subunit. In this context, Novavax recently reported a vaccine against COVID-19 with 89% efficacy using a saponin-based Matrix-M adjuvant.

#### 3.5.2. Viral Vector Vaccines

Viral vector vaccines use replication-deficient viruses designed to express the antigen's genetic sequence in host cells and improve immunogenicity induced by a cytotoxic T lymphocyte (CTL) response to kill virus-infected cells [94]. Concerning the SARS-CoV-2 vaccines using adenoviruses, adenovirus serotype 26 vector vaccine (Ad26.CoV2.S Johnson and ChAdOx AstraZeneca) (Figure 1) has shown promising results [92].

#### 3.5.3. mRNA Vaccines

mRNA vaccines are a promising alternative to traditional vaccine techniques due to their high potency, rapid development capability, and potential for low-cost manufacturing and safe delivery [95]. Concerning SARS-CoV-2 vaccines, lipid nanoparticles are used to safeguard the prefusion-stabilized S protein–encoding mRNA on its way to the intracellular area. The target protein is made by the host using the mRNA, which triggers a coordinated immune response [92]. Clinical outcomes from phase III trials for COVID-19 vaccines

made by Pfizer/BioNTech (New York City, NY, USA/Mainz, Germany) and Moderna (Cambridge, MA, USA) have considerably exceeded expectations, with vaccine efficacy rates approaching 95%, leading to a long-lasting germinal center B cell response that allow substantial humoral immunity [96–98].

## 4. Honey as a Co-Adjuvant of SARS-CoV-2 Infection Treatment

Honey and its related products are used as natural therapies for several health problems, including pulmonary and cardiovascular disorders, diabetes, hypertension, gastrointestinal tract disorders, edema, cancer, autophagy dysfunction, bacterial, viral, and fungal infections [99,100]. These biological properties are principally due to honey's potential antioxidant and anti-inflammatory activities, which may not only attenuate the oxidative damage induced by pathogens but also improve the immune system. In the specific case of viral infections, honey is employed due to its abilities to decrease acute inflammation by promoting an immune response [101,102]. In fact, honey can modulate the molecular targets involved in cellular signaling pathways, such as apoptosis and inflammation, as well as signaling cascades needed for virus replication and attachment to the host cells [103,104]. Varicella zoster [105], rubella [106], influenza [107], herpes simplex [108], respiratory syncytial virus [109], immunodeficiency virus [110], viral hepatitis A [110], are among the pathogens effectively inhibited by honey.

Honey is mainly composed of sugars and water, while minor compounds such as organic acids, amino acids, enzymes, phenolic compounds, vitamins, minerals, and antioxidants, are related with its biological properties [111–115]. As previously mentioned, the foraging activity of honeybees during honey production can lead to variations in honey composition, and are highly dependent of the plant species visited. Additionally, seasonal, environmental, and processing factors, as well as genus and bee species, may also influence honey composition and its biological effects [39,116]. Some research has suggested that many of the therapeutic properties of plants can be transmitted to honey, honey being a transport vector for the plant's medicinal properties [117]. Some honey varieties with similar botanical origins, and derived within close geographic proximity, have different biological properties because of the type and quantities of their bioactive components (Tables 1–3). Due to its composition and biological properties, honey and its related products are applied directly, or as components, in the treatment of several diseases. These products have proven their virucidal effect on several enveloped viruses such as HIV, influenza virus, herpes simplex, and varicella-zoster virus [105,107,118,119].

Among the different biological properties of honey, the most reported is its antioxidant activity. It is well known that antioxidants can prevent cell death by draining lymphocytes, which leads to antiviral action [114,120,121]. Corrêa and Rogero [122] observed that a polyphenol-rich environment stimulates the activation of the immune system and the mechanisms involved in tissue repair. Manuka honey, a monofloral honey derived from the nectar of the manuka tree (*Leptospermum scoparium*), an indigenous plant of New Zealand, has greatly attracted the attention of researchers due to its biological properties; especially its antimicrobial and antioxidant capacities [99,121]. The chemical composition of Manuka honey includes an abundant suite of polyphenols and other bioactive compounds such as glyoxal and methylglyoxal (MGO), methyl syringate, and leptosis [99,121]. Particularly, the $\alpha$-ketoaldehyde compound, MGO, can inhibit growth of the enveloped virus [107] and, together with the antiseptic effect of hydrogen peroxide, this aldehyde increases the antibacterial effect reported for this honey [113]. Combarros-Fuertes et al. [121] analyzed 16 different honey samples to select the best one for therapeutic purposes. The antioxidant activity, the main bioactive compounds and the phenolic profiles were determined. The samples exhibited great variability, with values ranging between 0.34 and 75.8 mg/100 g honey for ascorbic acid, 23.1 to 158 mg of equivalents of gallic acid/100 g honey for TPC and between 1.65 and 5.93 mg equivalents of catechin/100 g honey for the total flavonoid content (TFC). Forty-nine different phenolic compounds were detected. The concentration of the phenolic compounds and their phenolic profiles varied extensively

among samples (ranging from 1.06 to 18.6 mg/100 g honey). The same was observed regarding antioxidant activity. Although the samples with better combination of bioactive properties were avocado and chestnut honeys, all of them had great antibacterial activity.

Different in vitro and in vivo studies have supported that flavonoids can inhibit Angiotensin-Converting enzyme (ACE), the target binding receptor of the SARS-CoV protein, indicating that honey and related products could exhibit a marked activity for COVID-19 treatment [102,123–125]. However, these conclusions must be supported by experimental studies. Moreover, due to the presence of certain compounds, such as methyl-glyoxal (MGO), copper, ascorbic acid, flavonoids, nitric oxide, hydrogen peroxide, and its derivatives, honey can suppress viral growth by inhibiting viral replication and/or virucidal activity [107,120].

The bioactivity of honey against SARS-CoV-2 infection is mainly driven by three mechanisms (Figure 3): (a) direct virucidal properties; (b) regulation/boost of host immune signaling pathways, and (c) cure and/or improvement of comorbid conditions.

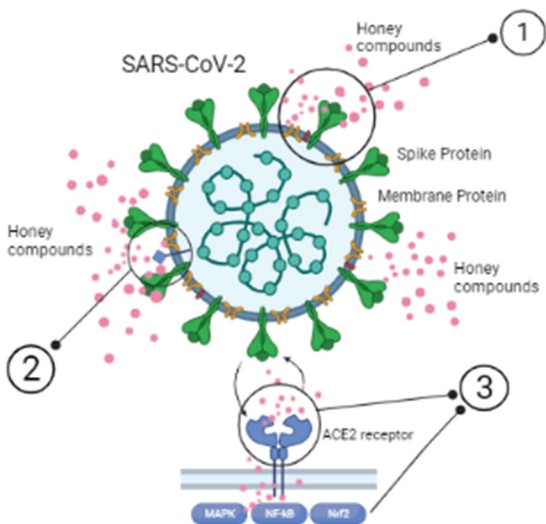

**Figure 3.** Possible mechanisms responsible for honey bioactivity against SARS-CoV-2: (1) altering the viral structure: interaction of honey and its major components with structural and/or non-structural proteins in the virus; (2) binding to target receptors on the virus; (3) interrupting membrane proteins (crucial for the viral attachment and entry into the host cells). Illustrations were made with BioRender.

### 4.1. Direct Virucidal Properties

The antiviral activity of honey may occur by different mechanisms. According to different authors, honey can inhibit viral infection by altering the structure of the surface protein, binding to target receptors on the virus, or interrupting the membrane proteins crucial for viral attachment and entry into the host cells [103,104]. Moreover, due to the presence of several compounds, such as MGO, copper, ascorbic acid, flavonoids, nitric oxide, hydrogen peroxide, and its derivatives, honey can suppress viral growth by inhibiting viral replication and/or virucidal activity [126]. Recently, a review focused on in silico, in vitro and clinical studies of honey and propolis on COVID-19, and highlighted the positive effect of flavonoids, such as, rutin, naringin, and quercetin, on treatment against SARS-CoV-2. The trace element copper is a well-known inactivator of viruses, while phenolic compounds, ascorbic acid and hydrogen peroxide inhibit viral growth by disrupting viral transcription, translation, and replication [126]. Regarding hydrogen peroxide, routine daily honey intake might provide protection against SARS-CoV-2 due to the biocidal effect of this reactive oxygen species, helping to clean the throat from virus particles. Moreover, the physicochemical properties of honey, namely pH (that ranges between 3.5 and 4.5), osmolarity, viscosity, and thickness, can also contribute to the antimicrobial effects reported [127].

### 4.2. Regulation/Boosting of Host Immune Signaling Pathways

Oxidative stress has a role in several pathological conditions, including neurological disorders, cancer, aging, endocrine illness, and pathogen infection. During virus invasion, oxidative stress induces inflammatory damage, with a consequently exacerbated immune response, the so-called cytokine storm [120]. Most SARS-CoV-2 patients showed elevated serum levels of C reactive protein (CRP), a marker of systemic inflammation [128]. The production of ROS and reactive nitrogen species (RNS) is generally counterbalanced by the action of antioxidant molecules or enzymes. The antioxidant effect of honey [101,104,129] is mainly correlated with the content of phenolic acids and flavonoids, sugars, proteins, amino acids, carotenes, organic acids, and water-soluble vitamins (vitamin B1, B2, B3, B9, B12, and vitamin C) [120,129].

Nuclear factor erythroid 2-related factor 2 (Nrf2)-dependent antioxidant gene expression is noticeably reduced in COVID-19 patients. Nrf2 stimulators may inhibit the replication of SARS-CoV-2 as well as related inflammatory gene expression, as demonstrated for honey [100,130]. As stated above, inflammatory processes are also a hallmark of SARS-CoV-2 infection. The inflammatory markers mitogen-activated protein kinase (MAPK) and the nuclear factor kappa B (NF-κB) induce the production of several other inflammatory factors, such as enzymes, cytokines, proteins, and cyclooxygenase-2 (COX-2), lipoxygenase 2 (LOX-2), CRP, interleukins (IL-1β, IL-6, and IL-10), tumor necrosis factor α (TNF-α), granulocyte–macrophage colony-stimulating factor (GM-CSF), vascular endothelial growth factor (VEGF), macrophage inflammatory protein (MIP1), MIP1A, MIP1B, platelet-derived growth factor (PDGF), and interferon-inducible protein 10 (IP-10) [118,122,128]. Recent in vitro and in vivo studies have demonstrated the anti-inflammatory mechanisms of honey. Hussein et al. [120] stated that honey decreases carrageenan-induced rat paw inflammation by attenuating NF-κB translocation to the nucleus, and by inhibition of the NF-κB degradation, with subsequent decrease of pro-inflammatory mediators COX-2 and TNF-α. Moreover, the same authors reported that honey also inhibited the production of the proinflammatory mediators nitric oxide (NO), prostaglandin E2 (PGE(2)), TNF-α, and IL-6, using the same animal model [112].

The attachment of SARS-CoV-2 to the cell surface occurs via interaction with the ACE-2 receptor, which is present in several cell surfaces, including lungs, heart, kidney, and arteries. Upon virus entrance, viral particles are recognizable by pattern recognition receptors (e.g., TLR3, TLR4, and TLR7). Inside human cells, the immune system is activated, including macrophages, natural killer cells, CD4+ and CD8+ T-cells, B-cells, neutrophils, and dendritic cells, aimed at the destruction of SARS-CoV-2 [120]. It has been reported that honey stimulates B-lymphocytes and T-lymphocytes in cell culture to multiply and activate neutrophils, which induce cytokine production, such as, IL-1, IL-6, and TNF-α and apalbumin 1 (AP-1) [131]. A variety of honeys have been associated with the increase of immune responses mediators [132,133]. Tonks et al. [134] discovered a 5.8 kDA component of Manuka honey that stimulates the production of TNF-α in macrophages via TLR4, such as blockade suppress honey-mediated immunomodulatory effects.

Besides all these phenomena, honey active phagocytosis and autophagy for pathogen clearance. Phagocytes are the first line of defense of the innate immune system, it being demonstrated that honey provides a supply of glucose essential for the "respiratory burst" of phagocytes (such as, polymorphonuclear neutrophils (PMNs) and peripheral blood mononuclear cells (MNCs)) [135]. On the other hand, autophagy is a highly conserved catabolic process that allows the cell to remove long-lived proteins, lipid, unwanted or damaged cells, as well as impurities, helping to maintain healthier cells. Different authors have reported that flavonoids, phenolic acids and MGO present in honey induce cell death by autophagy and by inhibition of the mTOR signaling pathway [136,137].

### 4.3. Cure and/or Improve Comorbid Conditions

Hyperglycemia has been found to be one of the causative risks factors of death in SARS-CoV-2 patients. Therefore, the hypoglycemic effect of honey previously reported

could be of huge importance for this condition [138,139]. Meo et al. [139] stated that honey could decrease fasting serum glucose as well as triglycerides and low-density lipoproteins (LDLs), increasing high-density lipoproteins (HDLs), the fasting C-peptide level and the 2-h postprandial C-level.

Besides hyperglycemia, myocarditis also contributes towards the final severe outcomes observed in SARS-CoV-2 patients [140]. Impaired lipid metabolism can lead to the elevation of total cholesterol, LDL, and triglycerides (TGs). Excessive ROS can attack the elevated LDL to gradually form atherosclerotic plaques that can block blood supply to the myocardium, inducing a hypoxic state that ultimately leads to the myocardial tissue necrosis [141].

Several in vitro, in vivo, and clinical trial studies have revealed positive honey effects against heart problems by improving the plasma lipid profile (i.e., reduced the level of very low-density lipoprotein (VLDL), LDL, TG, cardiovascular risk predictive index (CVPI), plasma cholesterol, TC, and HDL) [142], suppressing oxidation, attenuating the elevation of cardiac damage markers (e.g., creatine kinase (CK–MB) and cardiac troponin I) as well as aspartate aminotransferase (AST), lactate dehydrogenase (LDH), and alanine aminotransferase (ALT) [143]. Increasing activities of antioxidant enzymes, such as superoxide dismutase and glutathione peroxidase/reductase [144], and LDL resistance to oxidation [145], have been observed.

## 5. Conclusions and Future Perspectives

The biological properties reported in this review highlight the positive impact of honey as co-adjuvant for the treatment of COVID-19. The in vivo antiviral effects observed are based on the bioactive compounds present in honey, particularly polyphenols. Clinical trials already published support the molecular mechanisms that lead to honey's antiviral activity, particularly regulation and boosting of the host immune signaling pathways. Antibacterial activity and antioxidant content have a huge impact on comorbidities associated with infection by SARS-CoV-2. Molecular mechanisms behind the different properties reported for honey need to be thoroughly explored, particularly the link between specific bioactive compounds and their molecular targets inside the cell. Clinical trials enrolling COVID-19 patients should be considered in future.

**Author Contributions:** All authors contributed to the conceptualization of the article and to the writing—original draft preparation. S.S. and F.R. contributed to writing—review and editing. C.D.-M. to visualization, supervision, and project administration. All authors have read and agreed to the published version of the manuscript.

**Funding:** This research was funded by MTS/SAS/0077/2020—Honey+—New reasons to care honey from the Natural Park of Montesinho: A bioindicator of environmental quality & its therapeutic potential, and by the projects UIDB/50006/2020, UIDP/50006/2020, and LA/P/0008/2020, all supported by Fundação para a Ciência e a Tecnologia (FCT)/Ministério da Ciência, Tecnologia e Ensino Superior (MCTES).

**Data Availability Statement:** Not applicable.

**Acknowledgments:** We are thankful to the Fundação para a Ciência e a Tecnologia (Portugal) through project MTS/SAS/0077/2020—Honey+—New reasons to care honey from the Natural Park of Montesinho: A bioindicator of environmental quality & its therapeutic potential. This work was also financially supported by Portuguese national funds through projects UIDB/50006/2020, UIDP/50006/2020, UIDB/04033/2020 and LA/P/0008/2020, from the Fundação para a Ciência e a Tecnologia (FCT)/Ministério da Ciência, Tecnologia e Ensino Superior (MCTES). Francisca Rodrigues and Clara Grosso are thankful for their contracts (CEECIND/01886/2020 and CEECIND/03436/2020) financed by FCT/MCTES—CEEC Individual 2020 Program Contract. Juliana Garcia is grateful to FCT and BPI La Caixa Foundation, within the project titled 'AquaeVitae—Água Termal Como Fonte de Vida e Saúde"—"PROMOVE—O futuro do Interior" call 2020 and "AquaValor—Centro de Valorização e Transferência de Tecnologia da Água" (NORTE-01-0246-FEDER-000053), supported by Norte Portugal Regional Operational Programme (NORTE 2020) under the PORTUGAL 2020 Partnership Agreement, through the European Regional Development Fund (ERDF).

**Conflicts of Interest:** The authors declare no conflict of interest.

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
