# Peer review of "Honey as an Adjuvant in the Treatment of COVID-19 Infection: A Review"

_applsci, doi:10.3390/app12157800_

Round 1

Reviewer 1 Report

In attached file.

Author Response

       Sónia Soares

 REQUIMTE/LAQV- Polytechnic of Porto - School of Engineering

Rua Dr. António Bernardino de Almeida

4249-015 Porto

Portugal

Dr. Anca Pop, Prof. Dr. Felicia Loghin, Dr. Catalina Bogdan and Dr. Ionel Fizesan, PhD

Editors

We would like to thank you for giving us the opportunity to revise our manuscript, according to the reviewers' suggestions. We would also like to thank the 3 reviewers for their thoughtful comments on our manuscript “Honey as adjuvant on the treatment of COVID 19 infection – A Review” (applsci-1815387).

We have copied and pasted all reviewers’ comments below and address each one individually. Our point-by-point response to each of the reviewer’s comments is noted below in yellow font. Thank you for considering this manuscript for publication in Industrial Crops & Products.

We look forward to your response.

Yours sincerely,

Sónia Soares and on behalf of co-authors.

Reviewer 1

Authors of the submitted manuscript review possible beneficial effects of honey in the treatment of Covid-19. In doing so, however, they have dedicated the largest part of the paper to general information on honey, its application and composition in specimens of different geographical origin, and to Covid-19/SARS-CoV-2, again in a general sense (virus constituents, mode of action, host cell invasion, possible strategies to combat this virus, types of vaccines, drugs). The topic of the study starts practically from Chapter 2 and lasts 4 pages. There are thousands of articles on health-promoting effects of honey, as well as on the structure, activities and modes of counter-action concerning SARS-CoV-2. Authors of the present article should only briefly mention important factors, support them by references and focus their manuscript on the topic of the title – discuss which components of honey interfere with which cell mechanisms (possibly schematic presentation), which honey type and its component was shown to exhibit beneficial effect in which virus infection (more specific information than in the submitted text), in corona family specifically, perhaps proposition of the mode of action against SARS-CoV-2. As it is written now, the application of honey in specific virus infections is just listed and not elaborated as should be (for example, references 111-114, 123-126).

Response: We thank the reviewer feedback. We tried our best to revised the reviewer concerns and we hope now the manuscript is in conditions to be accepted for publication in Applied Sciences.

Altogether, authors should focus and avoid extensive generalization. The number of review articles on honey/Covid-19 is growing and I suggest authors to include/discuss 2 more in their paper: Dilokthornsakul et al. Potential effects of propolis and honey in COVID-19 prevention and treatment: A systematic review of in silico and clinical studies. J. Integr. Med. 2022, 20, 114-125, and Al-Hatamleh et al., Antiviral and Immunomodulatory Effects of Phytochemicals from Honey against COVID-19: Potential Mechanisms of Action and Future Directions. Molecules 2020, 25, 5017.

Response: We thank the reviewer advice and suggestion. The references were included in the manuscript.

Minor:

-English language needs thorough (professional) correction

Response: We thank the reviewer comment. The English and gramma were revised by a native speaker of our university. We hope now it is in line with the reviewer expectations.

-There are some technical mistakes

Response: We thank the reviewer advice. We revised all sentences regarding the scientific aspects, and we hope all is now correct.

-There are repetitions of the same information on several occasions (for example, that honey has anti-oxidant and anti-inflammatory activities) – repetitions should be deleted.

Response: We thank the reviewer comment. The information was completely revised, and we hope now the manuscript is in accordance with the reviewer expectation. 

Reviewer 2 Report

The tables have to be reedited.

Please provide bibliography for this paragraph -Different mild symptoms were registered, such as 120 fever, cough, and fatigue, to severe respiratory failure, similarly to general symptoms of 121 a viral infection and pneumonia. Additionally, the less common symptoms, such as diar- 122 rhea, myalgia, hemoptysis, and sore throats, were like those reported for other viruses, 123 namely Severe Acute Respiratory Syndrome (SARS) and Middle East Respiratory Syn- 124 drome (MERS).-

I would just suggest the authors also to include this articles in the review

DOI

10.3390/microorganisms8111704

DOI

10.2147/RMHP.S284557

DOI

10.3390/microorganisms9030525

DOI

10.2147/JIR.S282213

DOI

10.3390/microorganisms9040793

DOI

10.3390/antiox10060881

DOI

10.3390/children9020249

Author Response

Sónia Soares

 REQUIMTE/LAQV- Polytechnic of Porto - School of Engineering

Rua Dr. António Bernardino de Almeida

4249-015 Porto

Portugal

Dr. Anca Pop, Prof. Dr. Felicia Loghin, Dr. Catalina Bogdan and Dr. Ionel Fizesan, PhD

Editors

We would like to thank you for giving us the opportunity to revise our manuscript, according to the reviewers' suggestions. We would also like to thank the 3 reviewers for their thoughtful comments on our manuscript “Honey as adjuvant on the treatment of COVID 19 infection – A Review” (applsci-1815387).

We have copied and pasted all reviewers’ comments below and address each one individually. Our point-by-point response to each of the reviewer’s comments is noted below in yellow font. Thank you for considering this manuscript for publication in Industrial Crops & Products.

We look forward to your response.

Yours sincerely,

Sónia Soares and on behalf of co-authors.

Reviewer 2

The tables have to be reedited.

Response: We thank the reviewer comment. We revised the tables; however, it is not easy to include in the word file due to the size. If necessary, we can provide the tables in PDF.

Please provide bibliography for this paragraph - Different mild symptoms were registered, such as fever, cough, and fatigue, to severe respiratory failure, similarly to general symptoms of a viral infection and pneumonia. Additionally, the less common symptoms, such as diarrhea, myalgia, hemoptysis, and sore throats, were like those reported for other viruses, namely Severe Acute Respiratory Syndrome (SARS) and Middle East Respiratory Syndrome (MERS).

Response: We thank the reviewer comment. The sentence was removed since the manuscript was revised and deeply modified. We hope the reviewer understands.

I would just suggest the authors also to include these articles in the review

10.3390/microorganisms8111704I

10.2147/RMHP.S284557

10.3390/microorganisms9030525

10.2147/JIR.S282213I

10.3390/microorganisms9040793

10.3390/antiox10060881

10.3390/children9020249

Response: We thank the reviewer suggestion. However, the associated Editor informed us that these references should not be included. We hope the reviewer understand our position.

Reviewer 3 Report

A manuscript represents a critical, constructive analysis of the literature on the topic of honey as adjuvant on the treatment of COVID 19 infection. It does not only present an overview of relevant streams of thought in a topic covered, but also adds new insights on developments in the area, indicating what the open questions are.

However, undertaking a review of the related literature assessment is an important part of any discipline. It helps to maps and assesses the existing knowledge and gaps on specific issues which will further develop the knowledge base. Systematic literature review differs from traditional narrative reviews by adopting a replicable, scientific and transparent producers. It helps to collect all related publications and documents that fit our pre-defined inclusion criteria to answer a specific research question. It uses unambiguous and systematic procedures to minimize the occurrence of bias during searching, identification, appraisal, synthesis, analysis, and summary of studies. When the procedure is done properly and has the minimal error, the study can provide reliable findings and reliable conclusion that could help decision-makers and scientific practitioners to act accordingly. Well done procedure for the systematic literature review process is essential and it ensures that the work is carefully planned before the actual review work starts.

This is why the systematic reviews (like this one) must have a methods section. This section enables motivated researches to repeat the review. If for any reason, the authors would like to avoid a separate materials and methods section, then they should include some information about applied methods at the end of the introduction. The information should contain data sources (e.g., bibliographic databases), search terms and search strategies, selection criteria (inclusion/exclusion of studies), the number of studies screened and the number of studies included etc. Please provide this info to the readers.

Additionally, abstracts of scientific papers are sometimes poorly written, often lack important information. Although some journals still publish abstracts that are written as free-flowing paragraphs, most journals require abstracts to contain the usual sections like Background, Methods, Results, and Conclusions. The abstract of this manuscript acts as an introduction to the manuscript and misses to provide the readers with important info regarding most important findings and conclusions. Therefore, it needs to be completely rewritten.

Author Response

       Sónia Soares

 REQUIMTE/LAQV- Polytechnic of Porto - School of Engineering

Rua Dr. António Bernardino de Almeida

4249-015 Porto

Portugal

Dr. Anca Pop, Prof. Dr. Felicia Loghin, Dr. Catalina Bogdan and Dr. Ionel Fizesan, PhD

Editors

We would like to thank you for giving us the opportunity to revise our manuscript, according to the reviewers' suggestions. We would also like to thank the 3 reviewers for their thoughtful comments on our manuscript “Honey as adjuvant on the treatment of COVID 19 infection – A Review” (applsci-1815387).

We have copied and pasted all reviewers’ comments below and address each one individually. Our point-by-point response to each of the reviewer’s comments is noted below in yellow font. Thank you for considering this manuscript for publication in Industrial Crops & Products.

We look forward to your response.

Yours sincerely,

Sónia Soares and on behalf of co-authors.

Reviewer 3

A manuscript represents a critical, constructive analysis of the literature on the topic of honey as adjuvant on the treatment of COVID 19 infection. It does not only present an overview of relevant streams of thought in a topic covered, but also adds new insights on developments in the area, indicating what the open questions are.

Response: We thank the reviewer feedback. We tried our best to revised the reviewer concerns and we hope now the manuscript is in conditions to be accepted for publication in Applied Sciences.

However, undertaking a review of the related literature assessment is an important part of any discipline. It helps to maps and assesses the existing knowledge and gaps on specific issues which will further develop the knowledge base. Systematic literature review differs from traditional narrative reviews by adopting a replicable, scientific and transparent producers. It helps to collect all related publications and documents that fit our pre-defined inclusion criteria to answer a specific research question. It uses unambiguous and systematic procedures to minimize the occurrence of bias during searching, identification, appraisal, synthesis, analysis, and summary of studies. When the procedure is done properly and has the minimal error, the study can provide reliable findings and reliable conclusion that could help decision-makers and scientific practitioners to act accordingly. Well done procedure for the systematic literature review process is essential and it ensures that the work is carefully planned before the actual review work starts. This is why the systematic reviews (like this one) must have a methods section. This section enables motivated researches to repeat the review. If for any reason, the authors would like to avoid a separate materials and methods section, then they should include some information about applied methods at the end of the introduction. The information should contain data sources (e.g., bibliographic databases), search terms and search strategies, selection criteria (inclusion/exclusion of studies), the number of studies screened and the number of studies included etc. Please provide this info to the readers.

Response: We thank the reviewer for all comments and suggestion. We completely agree with the reviewer. The information was included in the manuscript (Specific information on the topic was collected from the literature available from search engines such as Google Scholar, PubMed, Science Direct, Scopus, and Web of Science for retrieving published data (from 2000 to 2022) using different combination of keywords i.e., COVID-19/SARS-CoV-2, honey, mechanism of action, immunomodu-lation/anti-inflammatory/antiviral, etc. The inclusion criteria limited to full text articles on pharmacological or therapeutic approaches for COVID-19 based on in-vitro, in-vivo and clinical trial reports.). We hope it is in accordance with the reviewer expectation.

Additionally, abstracts of scientific papers are sometimes poorly written, often lack important information. Although some journals still publish abstracts that are written as free-flowing paragraphs, most journals require abstracts to contain the usual sections like Background, Methods, Results, and Conclusions. The abstract of this manuscript acts as an introduction to the manuscript and misses to provide the readers with important info regarding most important findings and conclusions. Therefore, it needs to be completely rewritten.

Response: We thank the reviewer comments and suggestion. The abstract was revised. We hope the manuscript is now in conditions to be accepted by Applied Sciences. 

Round 2

Reviewer 1 Report

Authors have only partially taken into consideration my comments. As I have pointed previously, the review would sound much better if it focused on the topic - honey effects in relation to viral infection.

English still needs correction. 

Reviewer 3 Report

The authors have successfully addressed all the issues raised by the reviewers. The manuscript can be published as is.